# Patient Care in Community Pharmacies during the COVID-19 (SARS-CoV-2) Pandemic: Effectiveness of Post-Graduate Education and Further Training Courses on Revenues

**DOI:** 10.3390/ijerph20053774

**Published:** 2023-02-21

**Authors:** Francesca Baratta, Elena Folpini, Michele Ciccolella, Paola Brusa

**Affiliations:** 1Department of Drug Science and Technology, University of Turin, Via Pietro Giuria 9, 10125 Turin, Italy; 2New Line Ricerche di Mercato, Via Riccardo Lombardi 19/10, 20153 Milano, Italy; 3FarmaHiSkill Italia, Via Guelfa 5, 40138 Bologna, Italy

**Keywords:** community pharmacies, pharmacists, COVID–19, SARS-CoV-2, Italy, education, training courses

## Abstract

Thanks to their distribution throughout the territory and extended opening hours, community pharmacists are among the healthcare specialists most easily accessible to the public and often represent the first point of consultation both for the treatment of acute health conditions and, more generally, for health and therapy advice. The objective of the present study was to evaluate whether post-graduate courses/further training courses for pharmacists might influence the quality of patient management and care and, consequently, the satisfaction of the users who entered the pharmacy. We used the revenues of the pharmacies (Group A) in which these pharmacists are employed as a performance indicator. We compared the data for this group with the national averages for Italian pharmacies (Group B) and with those of a group (Group C) of selected pharmacies as similar as possible to the pharmacies in Group A based on a number of well-defined parameters. The comparison of revenues, year-on-year changes, and the average number of sales by the pharmacies in the three groups indicates that the pharmacies in Group A had the best performance, not only when compared with the national average but especially compared with the control group, specifically selected to make the comparison as significant as possible.

## 1. Introduction

Thanks to their distribution throughout the territory and extended opening hours, community pharmacists are among the healthcare specialists most easily accessible to the general public and often represent the first point of consultation both for the treatment of acute health conditions, particularly of minor entities, and, more generally, for health and therapy advice [1].

In recent years, the role of the community pharmacy has expanded significantly: from the simple dispensation of products to the provision of new services, patient management, and long-term care, in particular for chronic pathologies. The modern community pharmacy not only responds to the immediate needs of individual patients but also safeguards public health by raising awareness of healthy lifestyle choices, developing preventive healthcare, and encouraging proper adherence to pharmacological therapies [2,3]. Hence, the community pharmacy is an important structure that offers a range of services to protect and maintain the public’s health [4].

During the peak of the COVID-19 (Corona Virus Disease-19) pandemic, community pharmacies proved to be of even greater value as, having stayed open and accessible to the general public throughout the lock-down, they came to be seen as a vital source of products such as PPE (Personal Protective Equipment), but also a reference point for reliable science-based information regarding the unknown infection [5]. The same is true in the Italian context, where the essential role of community pharmacies is confirmed by the fact that the average number of customers entering a community pharmacy each day never decreased throughout 2020 and 2021, with an average of approximately 180 entries per day [5].

In a recent study [4], we analysed the relationship between the community pharmacy and its users during the various phases of the pandemic. The results of the survey, carried out in a number of Italian regions, showed that 70% of the respondents regarded the role of the pharmacy during the pandemic as absolutely essential, while 98% of subjects affirmed that they had received essential information regarding the pandemic, safety measures, and the new virus. These data are extremely relevant in that they demonstrate the positive impact that the community pharmacist can have in countering the spread of infodemia [6] (too much information, including false or misleading information during a disease outbreak), a growing phenomenon in the health field. In contrast, the two aspects identified as most disruptive by the respondents were the suspension of the services normally available in community pharmacies and the lack of direct contact with the pharmacist owing to the social distancing measures imposed to contain the pandemic. Concerning the future, in reply to the question of how the pharmacy could be improved following the events during the emergency phase of the pandemic, the most popular options chosen by respondents in percentage terms were the expansion of the range of services available in community pharmacies, and greater coordination between the pharmacist and the family physician, something which is not currently regulated by law in Italy. Regarding the complete digitalisation of pharmacies in the future, at least 50% of respondents strongly disagreed. Overall, the survey indicated that users generally have a positive opinion of community pharmacies, are strongly opposed to a complete digitalisation of the service, and strongly support face-to-face contact with the pharmacist.

After evaluating the overall satisfaction of Italian users with community pharmacies, we felt it was opportune to investigate whether this positive view was linked to the idea of the community pharmacy as a place in which to obtain assistance and care or was also influenced by the specific characteristics of individual community pharmacies. In particular, the objective of the present study was to evaluate whether post-graduate courses/further training courses for pharmacists might influence the quality of patient management and care and, consequently, the satisfaction of the users who entered the pharmacy, assuming that the basic degree-level education and training for pharmacists is uniform across the national territory.

## 2. Materials and Methods

To evaluate the effect of post-graduate education/further training on the activity of community pharmacists who had participated in such courses, we used the revenues of the pharmacies in which these pharmacists are employed as a performance indicator. These pharmacies were classified as Group A. We compared the data for this group with the national averages for Italian pharmacies (Group B) and with those of a group of selected pharmacies as similar as possible to the pharmacies in Group A on the basis of a number of well-defined parameters (Group C). The participants all took part on a voluntary basis.

In all the Group A pharmacies, at least one of the pharmacists had taken part in a post-graduate training course regarding scientific subjects such as patient management and selection of therapy, conformant to the current legislation on authorisation to prescribe, but also training on certain topics usually outside of the university curriculum. Specifically, we selected those pharmacies where staff had received training in management, psychology, and sociology. Courses with a minimum duration of at least six months and weekly attendance were taken into consideration. Moreover, the training courses had to include an examination for each subject and a final thesis. These kinds of training courses are not compulsory for the continuous professional training of pharmacists in Italy. Therefore, Group A comprised pharmacies where, to achieve a balance between professional ethics and business skills, the community pharmacist had studied business management, including such topics as budgeting and forecasting, preparing a balance sheet, financial planning and cost control, and marketing. The participating pharmacists had also taken part in a psychological and sociological training course with two aims: the creation of an effective team within the pharmacy after reorganising the duties and responsibilities of each staff member and the use of interpersonal skills with users from a communicative point of view.

The pharmacies in Group C were defined according to the technique of exact matching [7,8,9,10] based on a 6-month reference period immediately prior to the pandemic. For each pharmacy in Group A, a corresponding pharmacy was identified with similar observable characteristics. The objective was to prevent bias due to external factors which are not measurable, such as restrictions to entry (e.g., barriers to access, traffic restrictions, availability of parking) or the presence of competitors in the surrounding area (i.e., other points of sale authorised to sell Over The Counter—OTC-medicines, or toiletries), but might affect the performance of the business. The variables used in the matching process were the size of the pharmacy in terms of total sales volume; the proportion of sales of prescription medicine to total sales; the local territory; and the main characteristics of the local population: age, sex, education, and income. Group C pharmacists were asked to confirm that they had not attended a post-graduate/training course such as those described for Group A.

The statistical accuracy of the data relating to the pharmacies in Group A and those in Group C was calculated using the Mean Absolute Percentage Error (MAPE) [11].

The comparison of the revenues of the pharmacies in Group A, B and C was based on data gathered in the first two quarters of 2021 and, for comparison, the same periods in 2020. The successive quarters in 2021 and 2020 were excluded to eliminate the effect of SARS-CoV-2 testing. This was initially available only in a few regions in Italy and, moreover, administered only in pharmacies which met the requirements of sufficient space and the presence of qualified pharmacists, i.e., those who had completed training to administer COVID tests or vaccines.

## 3. Results

A total of 41 pharmacies in 10 Italian regions were included in Group A: 4 in Northern Italy (Piedmont, Liguria, Lombardy, and Veneto), 2 in Central Italy (Emilia Romagna and Lazio), and 4 in Southern Italy (Basilicata, Campania, Puglia, and Sicily). The geographical distribution of the pharmacies in Group A is reported in Table 1. The geographical distribution of the pharmacies in Group C is the same as those in Group A.

From the analysis of the revenues in the first two quarters of 2021 (Table 2), it is possible to note that the pharmacies in Group A had the highest average total revenues in both of the quarters in question—and, hence, the best performance—compared with the pharmacies in Group C and the pharmacies in Group B. The greatest difference in average revenues among the three groups of pharmacies can be observed in the “commercial” segment (OTC medicines and other categories of non-prescription products), both in terms of revenues and percentage change in revenue (Year Over Year comparison—YOY). It is important to point out that while the pharmacies in Group C have higher average total revenues than those in Group B, the percentage change in average total revenues is greater (−9.7% and −9.3% in the first quarter respectively, and +8.1% and +8.8% in the second quarter respectively) both for the commercial and the prescription segments (“prescription products”). In terms of revenues and change, the pharmacies in Group A were affected to a lesser extent by the pandemic. This is particularly evident in the first half of 2021, when there was a fall in revenues compared with the same period in 2020, a time in which cases of COVID-19 had not yet been recorded but, in any case, not as sharp as the fall in revenues in pharmacies in Group B or in Group C. Moreover, in the second quarter of 2021, the rebound in revenues in the pharmacies in Group A is more pronounced than in the other groups.

Observing the sales tables both in terms of volume (number of sales), and value (average value per sale), it is notable that the performance of the pharmacies in all three groups is similar when quantified by the total number of sales. In contrast, the average value of the sales in the “commercial” segment in significantly higher in the pharmacies in Group A. Table 3 reports the data from the second quarter of 2021; this is compared with the same period in 2020; both quarters were affected by the pandemic emergency. The data indicate that the average value per sale in the “commercial” segment in pharmacies in Group A was higher, although the number of sales was similar to that of pharmacies in both Group B and Group C: hence, the number of users entering the pharmacies in Group A is no greater, but spending on products by those users is higher, especially in the “commercial” segment.

For the pharmacies in Group A and those in Group C, a comparative analysis (Table 4) of the data for the prescription segment reveals no significant differences in revenues in the period between January–June 2021 (MAPE < 7%). This confirms that the service had little or no impact on the number of users entering the pharmacy. However, if we compare the changes in revenues in the two groups of pharmacies year on year, the table for the pharmacies in Group A is higher, especially in the therapeutic sectors for patients with chronic morbidities. Observing, for example, the cardiovascular system sector, and the nervous system sector, the two sectors generating the highest revenues, the difference in change in revenues (YOY) for the two groups of pharmacies amounts to approximately one percentage point.

When the same evaluation is carried out on the commercial segment for the pharmacies in Group A and those in Group C (Table 5), the data are even more significant as virtually all the sectors show a better performance, both for revenues and the change year-on-year. This is especially so for the self-medication sector: the negative impact on revenues as a result of the missing flu season in the autumn of 2020 [12] was almost completely offset by these increases for the Group A pharmacies, while all the other sectors show significant increases.

The sales of some commercial products with a certain ethical-social value can be analysed in greater detail, namely the prevention of diseases such as melanomas or sexually transmitted diseases. Regarding the dermo-cosmetics sector, and in particular sun cream products, it is interesting to note how, in the second quarter of 2021, the growth in average revenue and the number of items per sale (YOY) rose more significantly in the pharmacies in Group A (change in revenue +34.4%, items sold +30.3%) with respect to those in group C (change in revenue +33.6%, articles sold +28.6%). The contraceptive sector also saw a more positive increase in the pharmacies in Group A (change +9.2%, articles sold +4.4%) compared with the pharmacies in Group C (change (YOY) +7.9%, articles sold +0.7%).

In addition, turning to the prescription segment and examining specifically the performance of generic medicines, the year-on-year change in revenues for the half-year period January–June 2021 was more positive for the pharmacies in Group A with respect to those in Group C (+1.2% and −0.4%, respectively).

Veterinary products, too, both prescription and commercial segments, yielded similar results. In the second quarter of 2021, the revenues, the change (YOY), and the number of items per sale are more positive year on year in the pharmacies in Group A (revenues 16,488, change +13.2%, articles per sale +6.7%) compared with those in Group C (revenues 14,692, change +7.6%, items per sale +4.7%). The tables are similar when considering the prescription and commercial segments separately.

## 4. Discussion

The comparison of revenues, year-on-year changes, and the average number of sales by the pharmacies in the three groups indicates that the pharmacies in Group A had the best performance, not only when compared with the national average (pharmacies in Group B), but especially compared with the control group (pharmacies in Group C) specifically selected to make the comparison as significant as possible.

In the prescription segment, any bias on sales by the pharmacist can be ruled out as the current legislation in Italy authorises only a physician to prescribe therapies. Thus, for this sector, it is certainly noteworthy that the pharmacies in Group A have sales figures that are particularly positive for prescription medicines for chronic conditions. This is interesting if one considers that the volume of prescription medicines dispensed is directly linked to the number of users entering the pharmacy. Creating loyalty among users suffering from chronic conditions is therefore essential given that these users, who enter the pharmacy to fill a prescription, may also decide to purchase products for the commercial segment, and it is precisely these products that significantly boost per-sale value in the pharmacies in Group A compared with those in the other groups. Indeed, given an identical number of sales by the pharmacies in the three groups, the average value of sales from the commercial segment is generally higher in the pharmacies in Group A. Therefore, the number of users entering the pharmacy is no greater, but the value of sales per user is higher. Hence, while the organisation in the pharmacies in Group A did not lead to an increase in the number of users who make a purchase, it can certainly be affirmed that the pharmacist’s interaction with the user maximised the value of that interaction.

The fact is that revenues from the commercial segment grew rapidly in the pharmacies in Group A, not only in comparison with the pharmacies in Group B but also compared with the pharmacies in Group C; this must certainly be linked to a well-designed strategy involving the entire team of pharmacists in the pharmacy and not merely to rely on the effectiveness of any displays or merchandising. Furthermore, it is important to underline the fact that the growth in sales in the pharmacies in Group A also includes products such as sun creams and contraceptive barriers useful for the prevention of diseases, or the veterinary sector, which, given the high costs of veterinary products entirely at the expense of the purchaser, may lead to an interruption of the therapy or inappropriate use of human medicines given the lower cost. Encouraging the use of generic medicines has a social value as this means a significant saving for the patient, who, thanks to the community pharmacist, is educated in the use of these types of products, which are still under-used in Italy.

The skills and acumen required to create a business model similar to that in the pharmacies in Group A, able to absorb the economic effects of the pandemic, is the result of a synergy between a specifically designed training programme and its effective implementation through a working methodology which evolves into a model of governance of the pharmacy transcending merely intuitive management of the business. By governance, we mean the array of tools, rules, relations, processes, and company systems that are useful for achieving proper and efficient management of the business. This definition sums up the complexity of the pharmacy sector, which, in its organisational restructuring as a consequence of the pandemic, for the pharmacies in Group A, has focused attention on the application of a few fundamental concepts acquired in a post-graduate training programme for community pharmacists who work there.

In this context, there must be a shared vision among all the staff working in a determined community pharmacy on what features make it unique and distinguish it from other pharmacies; this must translate into measurable data in terms of quantity and quality. It is then necessary to define a clear and planned professional development programme for the pharmacists employed in a particular business, starting from the identification of strengths and weaknesses to help them respond effectively to the needs of their users. Being a small business no longer means it is unnecessary to adopt an organisational model in terms of roles, responsibilities, and duties. The functions of the staff members must be organised within a clear workflow which, not forgetting the profitability of the pharmacy, provides a range of services that ensure customer loyalty. Lastly, the definition of levels of production helps to create the proper balance between professional identity and entrepreneurial spirit.

All things considered, financially solid community pharmacies could more easily introduce services which are considered more professionally prestigious but are currently viewed as unprofitable, such as a compounding laboratory or long-term management of patients with chronic illnesses. The beneficial role of the community pharmacist has been demonstrated with various pathologies, but currently, at least in Italy, they are not remunerated systematically within the National Health system.

## 5. Conclusions

This study aimed to analyse the effects of post-graduate courses/further training courses and their influence on the field of community pharmacy. A group of pharmacies, where at least one pharmacist took part in a specific training course (Group A), was compared with a control group (pharmacies in Group C) and with the national average (pharmacies in Group B).

The revenues of the three groups of pharmacies were compared. In Italy, each patient freely chooses the pharmacy, and he can each time decide to go to a different pharmacy to obtain medicines or services; therefore, we considered it highly probable that a higher revenue is linked to more effective assistance for the patient.

The pharmacies in Group A showed better performances compared to the national average (Group B) and, above all, compared to the pharmacies in Group C. This was true both for the prescription segment and for the commercial sector.

The application of a precise working method has allowed Group A pharmacies to obtain excellent results despite the pandemic period, thanks to the development of a governance model that represents the synergy between the training course and its application. Such a model of governance can only develop based on specialist knowledge, which allows action based on planning rather than spontaneous intuition. This very approach, integrated into a training program, has enabled the pharmacies in Group A to adapt to and thrive in the turmoil of the pandemic and to steer a course which has yielded strong results and user satisfaction.

It would be desirable that the evolution of community pharmacy proceeds in parallel with the evolution of training courses. Nowadays, in Italy, training courses related to the recent evolutions of the pharmacist profession are only postgraduate ones. The training courses for the pharmacist vaccinator are a relevant example.

## Figures and Tables

**Table 1 ijerph-20-03774-t001:** Geographical distribution of pharmacies in Group A.

Region	Number of Pharmacies
Piedmont	20
Veneto	7
Lombardy	3
Puglia	3
Liguria	2
Sicily	2
Basilicata	1
Campania	1
Emilia Romagna	1
Lazio	1

**Table 2 ijerph-20-03774-t002:** Quarterly revenue (€) 2021 and change (Year Over Year comparison—YOY).

	First Quarter			Second Quarter		
	Group A	Group B	Group C	Group A	Group B	Group C
Average total revenue (€)	421,417	307,825	401,635	430,124	315,513	405,783
Change (YOY)	−6.3%	−9.3%	−9.7%	+10.9%	+8.8%	+8.1%
Average revenue “prescription” segment (€)	219,782	175,276	219,842	223,381	177,797	221,566
Change (YOY)	−8.8%	−9.8%	−10.0%	+7.7%	+7.8%	+7.9%
Average revenue “non-prescription products” segment (€)	201,635	132,550	181,793	206,743	13,7716	184,217
Change (YOY)	−3.5%	−8.6%	−9.3%	+14.7%	+10.2%	+8.4%

**Table 3 ijerph-20-03774-t003:** Analysis of sales data.

										Non-Prescription Products	
	Month	Working Days	Number of Receipts (Average)	Versus 2020	Pieces per Receipt (Average)	Versus 2020	€ per Receipt (Average)	Versus 2020	Medication under Prescription Incidence (%)	€ per Receipt * (Average)	Versus 2020
Group A	APR	25	4967	+15.0	2.66	−11.1	28.86	−6.4	57.5	21.72	−2.5
	MAY	25	5211	+16.5	2.61	−10.1	28.15	−3.8	58.1	21.49	+0.6
	JUN	24	5153	+8.7	2.65	−4.5	28.91	+0.7	57.8	21.85	+3.4
Group C	APR	25	4751	+15.0	2.78	−16.5	28.84	−7.8	57.8	20.65	−8.3
	MAY	25	4931	+15.7	2.73	−8.3	28.61	−1.3	57.7	20.62	−2.4
	JUN	25	4917	+11.1	2.78	−4.5	29.58	+1.8	57.8	21.10	+0.5
Group B	APR	25	4732	+16.3	2.77	−12.3	29.05	−6.7	57.7	20.76	−5.3
	MAY	26	4824	+13.9	2.75	−6.2	28.83	−1.2	57.5	20.61	−0.9
	JUN	25	4858	+9.7	2.78	−3.2	29.44	+0.2	57.0	21.02	+1.1

* Receipts which contain at least one product of the commercial area (non-prescription products): values reported within the table refer to the pure commercial component of the selected receipts.

**Table 4 ijerph-20-03774-t004:** Revenue and changes in the prescription sector.

	Group A		Group C	
	Average Revenue (€)	Versus 2020	Average Revenue (€)	Versus 2020
CARDIOVASCULAR SYSTEM	96,663	−1.4%	95,474	−2.3%
NERVOUS SYSTEM	66,877	+1.6%	69,624	−0.1%
RESPIRATORY SYSTEM	47,761	−11.0%	43,701	−13.1%
GASTROINTESTINAL	44,453	+2.0%	45,208	−0.7%
PAIN RELIEF	24,234	−2.1%	25,606	−1.3%
SYSTEMIC ANTIMICROBICS	20,919	−13.1%	19,949	−17.4%
METABOLISM	20,281	+0.1%	21,052	+8.8%
BLOOD AND BLOOD-FORMING ORGANS	17,611	−0.0%	17,630	+10.1%
UROLOGICALS	15,247	+0.1%	1400	+0.6%
GENITO URINARY SYSTEM	15,157	+7.4%	15,129	+4.1%

**Table 5 ijerph-20-03774-t005:** Revenues and trends in the “commercial” sector.

	Group A		Group C	
	Average Revenue (€)	Versus 2020	Average Revenue (€)	Versus 2020
SELF MEDICATION	218,932	−1.10%	201,462	−3.7%
DERMOCOSMETICS	70,132	6.40%	62,819	+3.4%
HEALTH AIDS	62,871	34.50%	47,330	+2.4%
DIET PRODUCTS	11,014	9.90%	11,049	+0.9%
VETERINARY PRODUCTS	16,488	13.20%	14,692	+7.6%
HOMEOPATHY	7940	−0.20%	5621	−3.5%

## Data Availability

All the data necessary for the correct understanding of the research are reported in the text. Non-aggregated data is available upon request.

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
