# Peer review of "Patient Care in Community Pharmacies during the COVID-19 (SARS-CoV-2) Pandemic: Effectiveness of Post-Graduate Education and Further Training Courses on Revenues"

_ijerph, 2023, doi:10.3390/ijerph20053774_

Round 1
Reviewer 1 Report
The authors aimed to investigate whether consumers satisfacion and quality of management and care in the pharmacy in CoVID pandemic is better if one or more pharmacists empoloyed completed postgraduate training /further training in the management, psychology or sociology.
The equivqlent of satisfaction is 6 month pharmacy revenue = quality indicator, in 2021 ( the first two quarters) compared to analog period of time the revenue in 2020.
The structure of income is described and compared
The question if the resuts can be explained only by training of one or more employees is not really answered but it is interesting to see that e.g. average income in sector of respitrtpory system medication in CoVID time dropped down in both groups ( A and C)
Reviewer 2 Report
This study examined whether post-graduate courses/further training for pharmacists could improve patient management and care and pharmacy users' satisfaction.
· Line 41: specify the pandemic name
· Line 43: give the full name of the abbreviation PPE
· Table 2: what is (YOY)? Use the full name once it is first mentioned
· Table 2: The unit for Average Total Revenue should be specified (e.g. dollars, euros, etc.)
· The manuscript contains spelling mistakes throughout and it is recommended to proofread and correct them.
· The conclusion should be revised to provide a clear summary of the findings and concrete recommendations for future actions.
· It is recommended to highlight the average hours received in professional development courses/workshops and to clarify the equivalency of continuing professional development (CPD) hours to post-graduate degrees.
· Clarify the inclusion and exclusion criteria for selecting the pharmacists/Pharmacies
· A separate table should be included to present the demographics of pharmacists, including their age and gender.
Reviewer 3 Report
A very interesting study, however, my main concern is that I am not sure how or why you correlated revenue with quality of care, this does not seem like an appropriate way of determining the level of care patients receive, especially as you are looking at the educational background of pharmacists.
Additionally, there was no description of the exact differences or characteristics of the pharmacists who were at these locations - how long have they been in practice, what kind of continuing education did they receive, etc.
I would encourage you to potentially re-think your title and objective if you are simply looking at revenue instead of actual "quality of care" or provide some information about a validated instrument that shows that quality = revenue and/or include more background as to why you chose to utilize this to determine quality.
Additional comments:
General
- Throughout the paper, you write the word group in both lower case and upper case when referring to the cohorts - please ensure that these are consistent throughout
Abstract
- You mention group A but you also have group B and group C in your article, so I would encourage you to list what these two groups are in the abstract as well
Introduction
- Line 54 - "These" should be "This"
Materials and Methods
- How did you decide on the pharmacises in Group A? What criteria was used? Did you specifically look at the types of training the pharmacists received?
- I am also not very clear as to the exact distinction between the three groups and if the data you looked at accounted for any duplication in available data between the groups
- Did you attempt to normalize the data - i.e did you take into consideration that some pharmacies may have more revenue simply because they have more clientele because of location, etc
-Educational difference between pharmacists in all groups.
Round 2
Reviewer 2 Report
All of the issues raised were adequately addressed by the authors. I recommend that the paper be accepted.